# Double-Blind Parallel Treatment Randomized Controlled Trial of Prebiotics’ Efficacy for Children Experiencing Severe Acute Malnutrition in Southern Punjab, Pakistan

**DOI:** 10.3390/children10050783

**Published:** 2023-04-26

**Authors:** Munazza Batool, Javeria Saleem, Rubeena Zakar, Sanaullah Iqbal, Ruhma Shahzad, Muhammad Salman Butt, Shahroz Haider, Florian Fischer

**Affiliations:** 1Department of Public Health, University of the Punjab, Lahore 54590, Pakistan; 2Department of Food Science and Human Nutrition, University of Veterinary and Animal Sciences, Lahore 54000, Pakistan; 3Bakhtawar Amin Medical and Dental College, Multan 60600, Pakistan; 4Institute of Public Health, Charité—Universitätsmedizin Berlin, 10117 Berlin, Germany

**Keywords:** RUTF, prebiotics, SAM, CMAM, malnourishment

## Abstract

The prevalence of malnutrition among children under five is alarmingly high in Pakistan. However, there are ready-to-use therapeutic foods (RUTF) available which may be used to treat children with severe acute malnutrition (SAM). This study aims to assess the efficacy of prebiotics as a synergistic additive to RUTF to enhance blood parameters and anthropometric measurements in children with uncomplicated SAM living in Southern Punjab, Pakistan. A double-blind parallel treatment randomized controlled trial was conducted on 204 children aged 6–59 months. Participants were randomized and allocated to the placebo (*n* = 102) or experimental arms (*n* = 102) in a 1:1 ratio. One group of children was provided with RUTF and 4 g prebiotics, while the other group was given RUTF and starch as a placebo. Participants recruited for both arms were given treatment for eight weeks, and then their biochemical and anthropometric outcomes were evaluated. A substantial difference between the mean weight, mid-upper-arm circumference, haemoglobin, haematocrit, platelet count, mean corpuscular volume, mean corpuscular haemoglobin, and albumin was revealed. During the two-month follow-up phase of the trial, children who were enrolled in the treatment group gained about 20% of their initial weight (pre-study mean weight = 5.44 ± 1.35 kg; post-study mean weight = 6.53 ± 1.45 kg). The analysis showed a significant difference (*p* < 0.005) between the control and treatment groups for MUAC and complete blood counts. Conclusively, supplementation with RUTF and prebiotics has proven to be an efficient, effective, and safe therapy for children suffering from SAM to improve their growth and development indicators and reduce the dangers of malnutrition in comparison to RUTF alone.

## 1. Introduction

Undernutrition is a rising cause of compromised health and untimely death in children under five, particularly in developing and underdeveloped countries. Malnutrition has serious and long-term effects on the developmental, social, economic, and medical aspects of individuals, families, communities, and countries at large [1,2]. Children who experience severe acute malnutrition (SAM) are at significant risk of lasting and adverse effects on their physical and mental health [3], which can negatively impact the economic and disease burden [4]. Sustainable Development Goals 1 and 2 address the eradication of poverty and hunger in susceptible populations. However, these issues still exist in the majority of low- and middle-income countries.

Compared to other developing countries, Pakistan ranks among the worst in the prevalence of malnutrition among children under five years of age [5], causing high rates of infant and under-five mortality rates. According to estimates by the Government of Pakistan, the infant mortality rate is 57 deaths per 1000 live births, while the under-five mortality rate is 65.2 deaths per 1000 live births. It is estimated that malnutrition is responsible for the deaths of approximately three million children under five years of age per annum globally [6,7], with poverty, economic instability and urbanization as the major factors contributing I hampering the health status of children [8].

To combat the risk of malnutrition in children under five and its associated long-term effects, a public health intervention named community-based management of severe acute malnutrition (CMAM) was initiated in Pakistan in 2011. The mainstay of this programme is the early detection of children suffering from SAM and the provision of basic healthcare and ready-to-use therapeutic food (RUTF) for the management of uncomplicated SAM in children. Research indicates the effectiveness of RUTF in treating such children across different parts of the world, including Pakistan [9,10]. The aggravating role of COVID-19 has caused an increase in the numbers of all forms of malnutrition among children which highlights the need of re-strategizing the actions for preventing and managing malnutrition among children [2]. However, the WHO has encouraged nutritional scientists and researchers to find different complementary therapies that could be utilized along with the administration of RUTF among children exhibiting uncomplicated SAM in order to enhance their developmental processes and weight gain [11].

Prebiotics are selectively fermentable dietary fibres, including galacto-oligosaccharides (GOS), fructo-oligosaccharides (FOS), and inulin, with positive and specific impacts on the microflora of the gut. Past studies have also shown that several prebiotics, including lactulose, are responsible for the management and control of intestinal microbiota by triggering the selective growth of lactic acid bacteria [12]. Furthermore, using the protection that prebiotics provide, local bifido-bacteria and lactobacilli can also control intestinal infections through a chain of potential mechanisms. These microorganisms secrete metabolic end-products that help reduce gut pH levels to such an extent that these pathogens are rendered completely ineffective.

Recent studies have revealed the gut microbiota to be a significant contributor to undernutrition [13]. Children who experience diarrhoea at an early age might experience impairment in the composition of their gut microbiota, which can result in continuous diarrhoea, leading to faltering growth and the reduced absorption of important nutrients. Therefore, prebiotics and probiotics can be used to support the gut microbiota and reduce diarrhoea and malnutrition [14]. In addition, some studies have shown that prebiotic and probiotic supplementation plays a significant role in improving growth and weight-gain parameters among infants who are born underweight [13,15]. The antimicrobial and anti-inflammatory properties of prebiotics and probiotics [16] contribute as a complementary factor that may enhance the response to standard therapy given to children experiencing uncomplicated SAM who are also struggling with the systemic inflammation and infections associated with it.

Previous studies have revealed a high prevalence of undernutrition among children living in circumstances of compromised hygienic and sanitary conditions, even when they do not suffer from diarrhoea or intestinal worms [17]. This is because the continuous absorption of faecal bacteria by children may result in a condition of environmental enteric distress, which in turn causes further complications, including inflammation of cells in the gut, poor villi functioning, leaking mucosa, and malabsorption that ultimately results in the body faltering [18]. Other factors that predict poor nutrition and health status in children include inadequate nutritional intake and economic, cultural, demographic, social, and environmental factors [17,19].

Different direct and indirect predictors of undernutrition among children under five years of age include various environmental factors, e.g., lack of hygiene, poor sanitation and quality of drinking water, poor level of education of mothers, lack of access to healthcare services, poor child healthcare practices, and food insecurity. According to a joint report by UNICEF, the WHO, and the World Bank [6], infections linked to diarrhoea can interfere with the absorption of nutrients, which may negatively influence children’s immunity [19,20] and can cause malnutrition. The treatment costs of diarrhoeal-associated infections may affect household and food budgets, and less food availability may result in insufficient nutrient intake and undernutrition among children under five years of age [21].

Therefore, using prebiotics as an additional component in the treatment of uncomplicated SAM in children can be a cost-effective approach to its prevention and treatment. Fortifying the microbiota with prebiotics could enhance energy and may protect against diarrhoea. To our knowledge, randomized controlled trials (RCTs) assessing the role of prebiotic supplementation in children with SAM have not previously been conducted in Pakistan. Therefore, clinical evidence is lacking and the topic of concern cannot be translated into clinical practice until scientific evidence has been generated, for which the present study has been established. The main focus of this study is to determine whether the addition of prebiotics to a standard RUTF will improve treatment outcomes (nutritional and clinical) in children affected by uncomplicated SAM.

## 2. Materials and Methods

### 2.1. Study Design and Setting

A double-blind parallel treatment randomized placebo-controlled clinical trial was conducted to determine whether the addition of prebiotics to a standard RUTF would improve treatment outcomes in children less than five years of age affected by uncomplicated SAM. This study was conducted in the southern Punjab region of Pakistan, as approximately 43% of the region’s population lives below the poverty line. The “National Programme for Family Planning and Primary Health Care” manages 52 functional outpatient treatment centres (OTPs) in the Dera Ghazi Khan (DGK) district of southern Punjab. The OTP centre of the teaching hospital in DGK was chosen to enrol children with SAM since this city connects all four provinces of Pakistan and is the primary hospital with an OTP centre where patients come for treatment. 

The chosen study location is underdeveloped, has inadequate socioeconomic housing due to poverty and illiteracy, is overcrowded, and has unsanitary living circumstances. DGK’s population comprises a tri-ethnic mix of natives (non-Baloch), Baloch, and Indian migrants (Muhajirs). The most underdeveloped region of southern Punjab is frequently impacted by floods and hill torrents. Children aged 6–59 months suffering from SAM without complications living in district DGK were selected as the population for the present study.

Predefined inclusion and exclusion criteria were used to recruit participants with a 1:1 allocation ratio to the two arms: Only clinically well children who were alert and had a good appetite were screened for possible recruitment after parental consent. Children with grade 1–2 bilateral oedema and a weight-for-height z-score of −3 or mid-upper-arm circumference (MUAC) of <11.5 cm who are otherwise clinically healthy, aware, and with a decent appetite are classified as children with SAM by the WHO. Children with any complications due to SAM (e.g., acute lower respiratory infection or hypoglycaemia, high pyrexia, hypothermia, anorexia, severe pitting oedema, or severe dehydration) were excluded from the study.

### 2.2. Sample Size

A minimum sample size of 158 participants (79 in each arm) was calculated using the formula given below.
$$n\;\left(per\;arm\right)=\frac{f_{\frac{\alpha }{2},\beta }x\left[p_{1}x\;\left(100-p_{1}\right)+p_{2}x\;\left(100-p_{2}\right)\right]}{\left(p_{2}-p_{1}\right)2}$$
where *p*_1_ is the success percentage in the control group, and *p*_2_ is the success percentage in the experimental group. We anticipated that 76% of children in the control arm would gain more than 15% of their baseline weight at 60 days. To detect a 16% absolute increase (to 92%) in the proportion of children gaining >15% weight at 60 days in the intervention arm, with 80% power at the 5% significance level, this was the bare minimum number of participants needed to complete follow-up.

This calculated sample size was increased to a total of 194 participants to account for attrition due to death and loss of follow-up by assuming a 25% international acceptable standard for the CMAM programme, >75% recovery rate, 15% default rate, and 10% death rate. Therefore, the total sample size in the present study, calculated and adjusted, was 194 participants (97 per arm).

### 2.3. Screening of Participants

Children between the ages of 6 and 59 months with a MUAC of <11.5 cm or weight-for-height z-score of −3SD were found in the community through a door-to-door survey. This was conducted with the agreement of the district chief executive officer utilizing lady health workers for the selected area. Children were inspected by these lady health workers using MUAC tape marked with a specific colour. To confirm SAM, they referred all children who met the requirements of the CMAM programme to an outpatient therapeutic programme for assessment by a competent nutritional staff nurse.

Once parents had agreed to the study, their children were then assessed further. In the first stage, children were evaluated for complications. A particular test was conducted to gauge the child’s hunger. The mother of the infant was given a RUTF sachet and instructed to give it to the child with lots of water. The child must consume at least one-third of this packet of RUTF to meet the requirement for a good appetite. The medical professional observed the child. The child was referred for inpatient treatment and excluded from the research if they did not consume at least one-third of the packet of RUTF (3 teaspoons equals 30 g) after three feeding attempts. This is a reliable method for assessing the child’s appetite; if they can eat half or more, this indicates a good appetite.

A detailed history was taken and a medical examination was conducted to assess the child’s health. Children suffering from high-grade fever, shortness of breath, severe vomiting, and oedema or who looked lazy, apathetic, or unconscious or had seizures were excluded from the study trial and referred to a stabilization centre for further treatment. Children with a history of recent watery diarrhoea associated with eyelid retraction, absence of urinary output, and tears with cold peripheries and lethargy were diagnosed with severe dehydration. They were also excluded from the study trial. Children were assessed for anaemia, and those whose palms were very pale, or gave a white look, were diagnosed with severe anaemia and were also not enrolled for the administration of RUTF [22] in the current trial.

Children were assessed for the presence of pitting oedema, and thumb pressure was applied to the ankle for three seconds. If a thumb impression remained for a few seconds on both feet, oedema was graded as mild (Grade 1, affecting both feet/ankles), moderate (Grade 2, affecting both feet, plus lower legs, hands, and lower limbs), or severe (Grade 3, generalized oedema including both feet, plus legs, arms, and face). Children suffering from severe pitting oedema (Grade 3) were excluded from the trial. Vital signs (temperature, pulse, and respiratory rate) of the child were also recorded: Children suffering from hypothermia or who had a high temperature (axillary temperature <35 °C or >39 °C, respectively) were excluded from the trial. Children with >50 breaths per minute, chest in-drawing, wheezing, or stridor were classified as having a likely acute lower respiratory infection and were not included in the trial. To check for hypoglycaemia, a heel- or finger-prick was performed by using a Dextrostix reagent strip. A glucose concentration of <3 mmol/L was considered hypoglycaemic, and such children were not enrolled in the trial.

### 2.4. Baseline (Pre-Test) Assessment

Children with uncomplicated SAM were then enrolled in the study and underwent baseline assessment. Sociodemographic and anthropometric measurements including MUAC and blood samples were collected from the randomized participants. Age-related information was recorded from antenatal records in cases of hospital-based delivery. For those children delivered at home, a maternal report was used to obtain age-related information. Weight and MUAC were measured by an outpatient-trained staff nurse. Recommended procedures and apparatus were used for measurements. Double measurements were taken, and in cases where there were differences between the two measurements, additional measurements were taken until an exact value was obtained. 

The children’s weight was assessed and recorded using UNISCALE [23] to the nearest 10 g, unclothed or in very light clothing. The scale was adjusted and calibrated to zero before each measurement. For infants and children who could not stand, the UNISCALE was used to measure the mother’s weight first. The mother was then handed the undressed baby/child while standing on the scales, and the combined weight of the mother and baby was measured. The difference between the two readings was used to calculate the baby/child’s weight.

The MUAC of each child was measured with colour-labelled MUAC tape to the nearest 0.1 cm, at the midpoint between the olecranon process and the acromion process. Children’s arms were measured with care, with the upper arm bent and exposed to the shoulder, and the lower arm resting transversely on the stomach while looking straight ahead. The elbow’s top bone tip and the top of the shoulder were identified, and the distance between the two tips was measured and divided by two to obtain the midpoint. The tape was applied across the arm at the designated midpoint and comfortably wrapped around the arm without being either too tight or too loose. To evaluate and classify the nutritional status of the children, MUAC was assessed. For data related to ethnicity and residence, the mother’s verbal history was taken into account.

### 2.5. Intervention

All children fulfilling the inclusion criteria were enrolled in the study for eight weeks and were allocated to either the placebo or experimental group with a 1:1 allocation ratio. Children blindly allocated to the experimental group were given RUTF with 4 g of prebiotic, while those in the placebo group were given RUTF with starch. Prebiotic galacto-oligosaccharides, Vivinal GOS powder, with a degree of polymerization between 2 and 10, were procured from Friesland Campina, Netherlands. The sachet contains 4 g of galacto-oligosaccharides. RUTF was procured from the IRMNCH programme of DGK Division, and starch for the placebo was taken from the market.

Galacto-oligosaccharides are one of the most extensively utilized prebiotic components for infant nutrition worldwide. Galacto-oligosaccharides are indigestible and have the advantages of promoting the development of bifidogenic bacteria, facilitating softer and more frequent faeces, enhancing natural defences, and enhancing mineral absorption. The limited use of this prebiotic in Pakistan and its positive effects motivated the researchers to utilize this as an intervention in their study to assess its positive health effects on children with uncomplicated SAM.

The number of RUTF sachets was determined according to the requirements and the child’s body weight. Two sachets were given for a child with a weight of 5–6.9 kg a day per week, three sachets were given for a child weighing 7–9.9 kg a day per week, and four sachets were given for a child with a weight greater than or equal to ten kg in a day/week. The training staff educated the parents about the importance of the RUTF and directed them on how to feed the child, and then the sachets were given to the parents. In the intervention arm of the trial, participants also received a daily dose of prebiotics, to be administered in such a way that prebiotics (4 g) were mixed with milk and water with the start of RUTF on the same day. Medicated sachets were packed in a clear cover and sealed by the manufacturer. In the control arm of the trial, participants received a placebo in a transparent sachet daily on the same day as RUTF started, and 4 g of placebo mixed with water and milk was also given to the participants. Medicated sachets were packed and covered with plastic by the registered pharmacist at the pharmacy of the Teaching Hospital, DGK District. Active and placebo-medicated sachets with unique identity numbers were stored in a sterile environment and a dry place for up to eight weeks, as recommended by the manufacturer.

### 2.6. Outcomes

The primary outcome measure for this study was the proportion of individuals gaining more than 15% of their baseline weight by the end of the two-month follow-up period. The proportion of individuals with anaemia at two months, and the mean serum levels of haemoglobin (Hb), haematocrit (HCT), mean corpuscular volume (MCV), mean corpuscular haemoglobin (MCH), platelets, and serum albumin at two months were the study’s secondary endpoints. Mortality and severe adverse events (SAEs) were considered to be safety endpoints in this study.

### 2.7. Randomization and Blinding

For the process of randomization and allocation to study groups, the statistician generated the random allocation sequence in an Excel spreadsheet. Equal numbers ranging from 1 to 200 were allocated to the placebo and active groups. No block size, stratification, or any other kind of limitation was applied. All the sachets of study medication, including both placebo and active study medication, were then labelled using the given sequence of numbers, along with the study number by the study pharmacy of the teaching hospital of DGK. In the CMAM programme, participants were enrolled and allocated consecutive identity numbers as per the sequence marked by the teaching pharmacy and then marked. An identification number was provided with the supplied sachets of placebo medication.

The sachets were prepared and transported to the study site within seven days. CMAM enrolment cards allocated to the already enrolled children in the hospital were assigned identity numbers. After this, the study medication prepared by the pharmacy department was given to each participant with the assigned identity number. A child with their whole identity number was provided with the sachet containing either the placebo (containing starch) or the treatment (oligosaccharides or prebiotics). Firstly, the sachet tagged with a particular identity number was verified and recorded on the CMAM enrolment card. Secondly, the identity number was confirmed again before giving the sachet to the particular child.

The staff nurses and health workers allocated the study participants and the parents/guardians were blinded from the study intervention. All the treatments, whether containing placebo or active medication, were in identical plastic sachets and had an identical texture and appearance.

### 2.8. Medications Permitted during the Study Period for Any Comorbidity

According to the CMAM protocol [22], the following medicine was allowed during the study period: a seven-day course of the antibiotic amoxicillin (60 g/kg/day/eight hourly) for children with a mild form of diarrhoea or other infections. Furthermore, a single dose of artesunate-amodiaquine combined therapy to treat children diagnosed with malaria was allowed, with syrup paracetamol if the child was suffering from fever (temp > 37.5 °C). Children suffering from dehydration due to diarrhoea could be given an oral rehydration solution made from ReSoMal (a rehydration solution for malnourished children).

### 2.9. Laboratory Methods

WHO guidelines on drawing blood and best practices in phlebotomy were adopted for a sample collection of 2–3 mL of whole blood in a Lavender-top (EDTA) tube labelled with a unique patient number. The drawn blood sample was sent to a local laboratory to record baseline data related to blood cell morphology, including haemoglobin (Hb g/dL), haematocrit (HCT %), mean corpuscular volume (MCV fl), mean corpuscular haemoglobin (MCH pg), platelets, and serum albumin (g/dL). The complete blood count analysis was performed using sysmax XN 1000, Japan, and the albumin analysis was performed on the Alinity Ci series, Abbot, IL, USA.

### 2.10. Follow-Up Assessment

Study participants were given CMAM enrolment cards. Parents of the children were advised to visit OTPs weekly for eight weeks for follow-up checkups. Lady health workers helped mothers or other family members to take the children for their routine visits to the study OTPs, where the nutritional supervisor evaluated the child’s overall health. Each child’s health was assessed weekly at the centre and was evaluated if there were any medical or nutritional complications, in line with standard practice and protocol. The need to refer children to the tertiary care hospital in case of any serious adverse events was also assessed. However, no such case was reported in any participant during the study period. The nurses gave the parents RUTF based on the child’s weight using a CMAM OTP chart. Parents were urged to follow up with their children for a health and nutritional status check on the seventh day. The benefits of feeding a child a healthy diet were also discussed with the mother. During the trial duration in the outpatient centres, parents were encouraged to bring their child to the centre for any ailment, and all treatments were free for the study participants. All participants 3 mL blood samples were also taken at the two-month follow-up for biochemical analysis and were sent to the local lab for the blood-cell morphology calculations. Developmental and physical health examinations and anthropometry were also performed again after two months: CMAM health workers who did not know the allocation procedure assessed the anthropometric measurements.

### 2.11. Statistical Analysis

SPSS version 25 was used for data entry, cleaning, and analysis. Frequencies, percentages, and mean ± standard deviation (SD) were calculated to demonstrate the descriptive analysis. Furthermore, the skewness of the data was checked. All the quantitative variables were normally distributed; paired sample *t*-tests were used to evaluate the difference between the outcome measures in the treatment group before and after the treatment, and independent sample *t*-tests were used to evaluate the difference between the outcome measures for each arm of the trial, i.e., placebo and treatment. All tests were applied at a 95% confidence interval (CI), considering a *p*-value < 0.05 as a significant value. No interim analysis was planned for this study.

### 2.12. Ethical Considerations

The Advanced Study Research Board at the University of Punjab approved the study (reference D/No 150/Acad). Written informed consent was obtained from all parents of children enrolled in the study. The study is registered in ClinicalTrial.gov under trial number NCT05390437.

### 2.13. Timeline of the Study

Children were recruited for the study trial from November 2020 to September 2021. The follow-up period was completed in December 2021, followed by data entry, cleaning analysis, and report writing.

To enrol 97 participants in each arm, a total of 256 children were screened for eligibility criteria. A few extra participants were recruited for each arm to compensate for cases of attrition due to loss of follow-up. The diagram below shows 107 and 106 participants recruited in the control and prebiotic arms, respectively. However, the parents of two children in the control arm and the parents of three children in the prebiotic arm withdrew their written consent. Additionally, three children from the control group and one from the prebiotic arm were lost in the follow-up. In the final data analysis, 204 children were included, 102 in each arm (Figure 1).

## 3. Results

### 3.1. Sociodemographic Characteristics

The majority of children were male (52.9%), aged up to 12 months (51.0%), Punjabi (60.3%), and resided in rural areas (63.7%). The precise frequencies and percentages are provided in Table 1.

### 3.2. Pre- and Post-Comparison of Growth Parameters for Treatment Group

A repeated measure design paired sample *t*-test was used to evaluate the effects of treatment with RUTF + prebiotics on different growth measures (weight, MUAC, HB, HCT, etc.) and to check whether the treatment is efficient. The analysis revealed that for pair 1, the weight of children before and after treatment (*p* < 0.001), pair 2 MUAC of the children (*p* < 0.001), pair 3 Hb of the children (*p* < 0.001), pair 6 MCV of the children (*p* < 0.001), pair 7 MCH of the children (*p* < 0.001), and pair 8 albumins of the children (*p* < 0.001), the effect of treatment was significant. Furthermore, the mean difference for all variables in the paired sample *t*-test was negative (weight = −0.77696; MUAC = −0.71176; Hb = −0.80490; HCT = −8.96912; platelets count = −5.50980; MCV = −3.21912; MCH = −1.98873; albumin = −0.32549), which indicates that the mean value for all variables had increased post-intervention, i.e., after supplementation with RUTF + prebiotics (Table 2).

### 3.3. Differences between Control and Treatment Group

The effect of RUTF and RUTF with prebiotics on the growth and development indicators of children with uncomplicated SAM was analysed. The analysis showed improvement in the growth indicators of children enrolled in both the treatment and control groups, including their mean weight, MUAC, HB count, HCT count, platelet count, MCV count, MCH count, and albumin. However, the mean growth parameters were significantly more improved in the treatment group compared to the control group (Table 3).

Before the trial, there were no significant differences in the mean weight of the children in the two groups. However, the analysis revealed that, at the end of the trial, there was a significant difference, with the mean weight of the children in the treatment group being 6.53 kg and that of the control group being 5.89 kg. Children enrolled in the treatment group gained approximately 20% of their baseline weight during the two-month follow-up period of the trial (pre-trial mean weight = 5.44 ± 1.35; post-trial mean weight = 6.53 ± 1.45). However, children enrolled in the control group gained 8.5% of baseline weight (pre-trial mean weight = 5.43 ± 1.55; post-trial mean weight = 5.89 ± 1.72). This highlights the significant role of RUTF with prebiotics on the weight of children with uncomplicated SAM (*p* = 0.005) (Table 3).

Similarly, the analysis revealed a significant difference between the MUAC (*p* = 0.001), HB (*p* < 0.001), HCT (*p* = 0.006), platelet count (*p* = 0.031), MCV (*p* < 0.001), MCH (*p* < 0.001), and albumin (*p* = 0.009) of children enrolled in the treatment and control groups post-trial, with children enrolled in the treatment group having larger mean MUAC (mean count = 10.80), HB count (mean count = 10.57), HCT count (mean count = 46.53), MCV (mean count = 74.35), MCH (mean count = 25.31), and albumin (mean count = 3.98). It can be concluded that, although RUTF has a role in improving some of the growth parameters in children with uncomplicated SAM when RUTF is combined with prebiotics, more significant positive results can be achieved. The detailed analysis of mean values pre- and post-trial among the control and treatment groups with their significant *p*-values is given in Table 3.

### 3.4. Adverse Events

No actual or suspected adverse events were reported or arose during the trial duration.

## 4. Discussion

Considering the dire effects of malnutrition among children, this study focuses on the role of RUTF individually and RUTF with prebiotics in improving the growth indicators of children with uncomplicated SAM. Likewise, past studies conducted on SAM among children revealed a high frequency of malnutrition among children residing in rural settings [24]. Our study also revealed that the majority of participants resided in rural areas (63.7%). The reason behind the high frequency of malnutrition among children in rural areas could be compromised hygiene and sanitation conditions.

The results of this study showed an improvement in the growth indicators of children enrolled in both groups, i.e., the treatment and control groups, including their mean weight, MUAC, HB count, HCT count, platelet count, MCV count, MCH count, and albumin level. However, the mean growth parameters were significantly more improved in the RUTF + prebiotics group compared to the RUTF-only group. In agreement with the findings of the present study, past studies on prebiotic supplementation have also revealed the significant role of prebiotics in improving growth and weight-gain parameters among infants, especially those who were born underweight, as well as indicating positive outcomes on the gut [25]. Likewise, some studies have also revealed significant improvements in Hb, HCT, and white blood cell levels among children suffering from SAM after prebiotic supplementation. On the other hand, Nakamura and colleagues [26] claimed to have found no significant effect of prebiotics on the growth indicators of children suffering from malnutrition. However, the fact cannot be neglected that, in that study, supplementation with an isotonic solution of prebiotics was administered individually, whereas, in most of the studies, prebiotics were administered in combination with other conventional therapies.

The findings of our study support the narrative that the antimicrobial and anti-inflammatory properties of prebiotics contribute as a complementary factor that results in the enhancement of response to standard therapy of RUTF given to children with uncomplicated SAM. Indigestible carbohydrates, such as prebiotics, have the ability to significantly alter the composition and activity of the intestinal microbiota, which may have impacts on the GI system. These prebiotics normally do not hydrolyse the polymer bonds in the human intestines, enabling them to endure small-intestinal digestion and pass through the colon intact, where they are fermented by healthy bacteria such as Lactobacilli and Bifidobacteria [27].

Malnutrition has detrimental and protracted repercussions on the social, economic, and medical facets of individuals, families, communities, and nations as a whole. The use of prebiotics (GOS) would become a relatively cost-effective intervention for SAM when compared to DALY, mortality, doctors’ visits, and missed school days. In a randomized controlled trial involving 111 children with median ages of 37 and 43 months, the cost-effectiveness and benefits of symbiotics in the treatment of children with acute infectious gastroenteritis were compared for the symbiotic (N = 57) and placebo (N = 57) groups. The cost of additional medications and overall healthcare was computed for this investigation. It was discovered that the symbiotic group’s median diarrhoeal duration was less than that of the placebo group. The placebo group received noticeably more prescriptions for antibiotics, antipyretics, and antiemetics than the symbiotic group did. In the placebo group, the calculated cost of additional medication was 4.04 euros per patient, compared to 1.13 euros per patient in the symbiotic group. Additionally, more frequent doctor’s consultations led to greater care costs per patient of 14.41 euros, compared to 10.74 euros in the symbiotic group. With an increase in symbiotic supplementation, the price of additional consultations and other medications was cut by 25% [28].

The main strength of our study is that, to the best of our knowledge, it is the first RCT conducted in Pakistan to evaluate the role of prebiotic (GOS) supplementation in combination with RUTF on the nutritional outcome of children with uncomplicated SAM. In addition, the data and anthropometric measures were collected by trained staff to minimize measurement bias and avoid missing data. Lastly, the study medication’s effect was observed to allow for 100% adherence. Although this study is one of a kind, there are some gaps that need to be mentioned here. On the one hand, the study was conducted in a single district of the South Punjab region, i.e., DGK, where a significant portion of people lives below the poverty line. Due to the homogeneous living conditions of the children, the results of the study may not be a good representation of children of the same age group across the country. Additionally, the follow-up duration of the study was short, highlighting only the short-term benefits of the treatment on the anthropometric measures and health indicators of children with uncomplicated SAM, and does not take into account any long-term benefits or the sustainability of the efficacy and effectiveness of the treatment. Therefore, there is a need to conduct future trial-based studies with a more extensive scope and a longer follow-up period, focusing on children with both complicated and uncomplicated SAM.

## 5. Conclusions

Conclusively, supplementation with RUTF and prebiotics has proven to be an efficient, effective, and safe therapy for children suffering from uncomplicated SAM. It was shown to improve their growth and development indicators and reduce the dangers of malnutrition. However, further longitudinal studies with longer follow-up periods are desirable to further evaluate the effect of RUTF and prebiotics on children’s growth and sustainability.

## Figures and Tables

**Figure 1 children-10-00783-f001:**
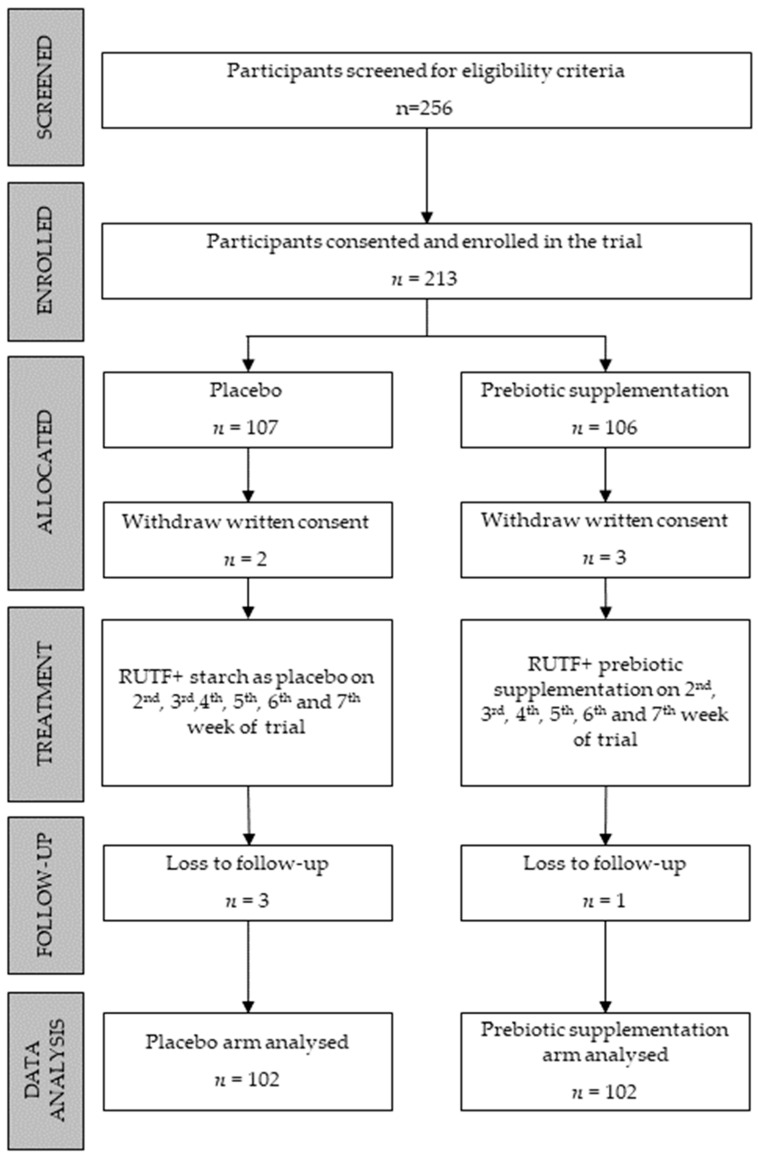
Flow diagram.

**Table 1 children-10-00783-t001:** Comparison of sociodemographic characteristics in the treatment and control group.

Sociodemographic Characteristics		Treatment *n* (%)	Control *n* (%)	Total
Gender (*n* = 204)	Male	51 (25%)	57 (27.9%)	108 (52.9%)
	Female	51 (25%)	45 (21.1%)	96 (47.1%)
Age (*n* = 204)	Up to 12 months	55 (27.0%)	49 (24.0%)	104 (51%)
	13–24 months	32 (15.7%)	39 (19.1%)	71 (34.8%)
	25–36 months	12 (5.9%)	9 (4.4%)	21 (10.3%)
	37–48 months	2 (1.0%)	3 (1.5%)	5 (2.5%)
	49–60 months	1 (0.5%)	2 (1.0%)	3 (1.5%)
Ethnicity (*n* = 204)	Baloch	28 (13.7%)	40 (19.6%)	68 (33.3%)
	Migrant	7 (3.4%)	6 (2.9%)	13 (6.4%)
	Native (Punjabi)	67 (32.8%)	56 (27.5%)	123 (60.3%)
Resident (*n* = 204)	Rural	61 (29.9%)	69 (33.8%)	130 (63.7%)
	Tribal	6 (2.9%)	7 (3.4%)	13 (6.4%)
	Urban	35 (17.2%)	26 (12.7%)	61 (29.9%)

**Table 2 children-10-00783-t002:** Pre- and post-comparison of the growth parameters of uncomplicated SAM children for the treatment group (RUTF + prebiotics) (*n* = 102).

Outcome Measures			Paired Differences					*p*-Value *
		Correlation	Mean Difference	SD	Std. Error Mean	95% CI of the Difference		
						Lower	Upper	
Pair 1	Pre-weight—Post-weight	0.893	−0.77696	0.73048	0.05114	−0.87780	−0.67612	<0.001
Pair 2	Pre-MUAC—Post-MUAC	0.801	−0.71176	0.85539	0.05989	−0.82985	−0.59368	<0.001
Pair 3	Pre-HB—Post-HB	0.504	−0.80490	1.38057	0.09666	−0.99549	−0.61432	<0.001
Pair 4	Pre-HCT—Post-HCT	−0.039	−8.96912	72.28303	5.06082	−18.94764	1.00941	0.078
Pair 5	Pre-platelets count—Post-platelets count	0.816	−5.50980	56,786.822	3975.872	−8390.283	7288.32299	0.890
Pair 6	Pre-MCV—Post-MCV	0.712	−3.21912	4.17567	0.29236	−3.79556	−2.64267	<0.001
Pair 7	Pre-MCH—Post-MCH	0.678	−1.98873	2.47062	0.17298	−2.32979	−1.64766	<0.001
Pair 8	Pre-Albumin—Post-Albumin	0.503	−0.32549	0.53860	0.03771	−0.39984	−0.25114	<0.001

* *p*-value calculated using paired sample *t*-test at 95% CI.

**Table 3 children-10-00783-t003:** Difference between the outcomes of children enrolled in the control (*n* = 102) and treatment (*n* = 102) group.

Outcome Measures	Groups	Mean ± SD	*t*-Test	*p*-Value *
Pre-weight	Treatment	5.44 ± 1.35	0.044	0.965
	Control	5.43 ± 1.55		
Post-weight	Treatment	6.53 ± 1.45	2.842	0.005
	Control	5.89 ± 1.72		
Pre-MUAC	Treatment	9.69 ± 1.20	−0.822	0.412
	Control	9.84 ± 1.39		
Post-MUAC	Treatment	10.80 ± 1.28	3.316	0.001
	Control	10.16 ± 1.45		
Pre-HB	Treatment	8.85 ± 1.28	−2.967	0.003
	Control	9.41 ± 1.37		
Post-HB	Treatment	10.57 ± 1.18	6.854	<0.001
	Control	9.32 ± 1.37		
Pre-HCT	Treatment	27.57 ± 4.43	−2.775	0.080
	Control	29.25 ± 4.11		
Post-HCT	Treatment	46.53 ± 102.19	1.757	0.006
	Control	28.56 ± 3.94		
Pre-platelets count	Treatment	280,234.00 ± 89,078.95	−0.526	0.600
	Control	287,502.00 ± 105,696.33		
Post-platelets count	Treatment	271,500.00 ± 60,080.00	−2.174	0.031
	Control	296,230.00 ± 96,572.14		
Pre-MCV	Treatment	68.99 ± 5.95	−1.411	0.160
	Control	70.16 ± 5.76		
Post-MCV	Treatment	74.35 ± 4.48	4.519	<0.001
	Control	71.31 ± 4.99		
Pre-MCH	Treatment	22.33 ± 3.33	−0.028	0.978
	Control	22.35 ± 3.24		
Post-MCH	Treatment	25.31 ± 2.55	5.130	<0.001
	Control	23.38 ± 2.79		
Pre-Albumin	Treatment	3.59 ± 0.52	1.045	0.298
	Control	3.51 ± 0.51		
Post-Albumin	Treatment	3.98 ± 0.470	2.648	0.009
	Control	3.77 ± 0.63		

* *p*-value calculated using independent sample *t*-test at 95% CI.

## Data Availability

Data is available from corresponding author upon reasonable request.

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
