# Peer review of "Double-Blind Parallel Treatment Randomized Controlled Trial of Prebiotics’ Efficacy for Children Experiencing Severe Acute Malnutrition in Southern Punjab, Pakistan"

_children, 2023, doi:10.3390/children10050783_

Round 1
Reviewer 1 Report
This study is very relevant in the field of nutrition. Authors should remember to include statistical inferences in the abstract section too. Should also work on English language and improve flow of ideas throughout the manuscript.
Author Response
>This study is very relevant in the field, of nutrition. Authors should remember to include statistical inferences in the abstract section too. Should also work on English language and improve flow of ideas throughout the manuscript.
The authors have acknowledged and complied with the reviewer’s comments and have mentioned the statistical inferences in the abstract section.
Section: Abstract
Lines 27-30: During the 2-month follow-up phase of the trial, children who were enrolled in the treatment group gained about 20% of their initial weight (pre-mean weight = 5.44 ± 1.35 kg; post mean weight = 6.53 ± 1.45 kg). The analysis showed a significant difference (p < 0.005) between the controlled and treatment groups for MUAC and complete blood counts.
Reviewer 2 Report
This study is of great interest and the protocol is rigorous. There are, however, some issues:
1. In the introduction prebiotics has to be defined (and not in the discussion) and probiotics as well.
2. In the methods section, please give the rationale for supplementing with prebiotics and not probiotics, and for choosing this specific prebiotic.
3. Why the pre-post changes were not compared between control and treatment group?
4. The mode of action of prebiotics should be discussed in more depth, even if it is only hypothesized.
5. The first three paragraphs of the discussion belong to the introduction.
6. The financial impact of this supplementation is another discussion point.
7. What was the percentage of complicated vs uncommplicated SAM in the study site?
8. Line 56-7: Provide the reference for the WHO recommendation.
9. Line 70: 'Systematic infection (or systemic?)' is a complication of SAM and therefore it does not pertain to uncomplicated SAM.
10. Table 2: 'Pair' is not needed.
Author Response
|
Reviewer 2 Comment # 01
In the introduction, prebiotics has to be defined (and not in the discussion), and probiotics as well. |
The authors have acknowledged and complied with the reviewer’s comments and have made the suggested changes as follows, Lines: 460-468 have been added to the introduction section Lines 64-72 |
|
Reviewer 2 Comment # 02 In the methods section, please give the rationale for supplementing with prebiotics and not probiotics, and for choosing this specific prebiotic. |
The authors have acknowledged and complied with the reviewer’s comments and have added the following information in the methodology section.
Line 237-243: Galacto-oligosaccharides are one of the most extensively utilized prebiotic components for infant nutrition worldwide. It contains indigestible galacto-oligosaccharides and has the advantages of promoting the development of bifidogenic bacteria, facilitating softer and more frequent faeces, enhancing natural defenses, and enhancing mineral absorption. The limited use of this prebiotic in Pakistan and its positive effects motivated the researchers to utilize this as an intervention in their study to assess its positive health effects on uncomplicated SAM children. |
|
Reviewer 2 Comment # 03 Why the pre-post changes were not compared between the control and treatment groups? |
The authors have acknowledged the reviewer’s comments and believe that the difference between the outcomes of children enrolled in the control (n=102) and treatment (n=102) groups was compared in Table 3. |
|
Reviewer 2 Comment # 04 The mode of action of prebiotics should be discussed in more depth, even if it is only hypothesized. |
The authors have acknowledged and complied with the reviewer’s comments and have made the suggested changes as follows
Line 479-484: Non-digestible carbohydrates, such as prebiotics, have the ability to significantly alter the composition and activity of the intestinal microbiota, which may have impacts on the GI system. These prebiotics normally do not hydrolyze the polymer bonds in the human intestines, enabling them to endure small-intestinal digestion and pass through the colon intact, where they are fermented by healthy bacteria like Lactobacilli and Bifidobacteria.
Reference: Pier, M., Guarino, L., Altomare, A., Emerenziani, S., Rosa, C. Di, Ribolsi, M., Balestrieri, P., Iovino, P., Rocchi, G., & Cicala, M. (2020). Mechanisms of Action of Prebiotics and Their E ff ects on Gastro-Intestinal Disorders in Adults. Nutrients, 12(4), 1037.
|
|
Reviewer 2 Comment # 05 The first three paragraphs of the discussion belong to the introduction. |
The authors have acknowledged and complied with the reviewer’s comments and have made the suggested changes as follows, Line 418-424: Is deleted from the discussion section. Lines 432-451 have been added to the introduction section lines 86-105. |
|
Reviewer 2 Comment # 06 The financial impact of this supplementation is another discussion point. |
The authors have acknowledged and complied with the reviewer’s comments and have made the suggested changes in the discussion section as follows;
Line 485-500: Malnutrition has detrimental and protracted repercussions on the social, economic, and medical facets of people, families, communities, and nations as a whole. The use of prebiotics can prove to be cost-effective and the cost-effectiveness ratio can be defined as the estimated cost of prebiotics divided by the estimated public health benefits ex-pressed in Pakistani rupees per avoided disability-adjusted life years ( DALY). The use of prebiotic (GOS) would become a relatively cost-effective intervention for SAM when compared to DALY, mortality, doctors’ visits and missed school days. |
|
Reviewer 2 Comment # 07 What was the percentage of complicated vs uncommplicated SAM in the study site? |
The authors have acknowledged the reviewer’s comment. In Pakistan, the prevalence of malnutrition among under-five children is alarming including in the DG Khan district. A prevalence of 2-3% of complicated SAM children and 8% of uncomplicated SAM children was estimated at the study site. |
|
Reviewer 2 Comment # 08 Line 56-7: Provide the reference for the WHO recommendation. |
The authors have acknowledged and complied with the reviewer’s comments
Reference: Guideline: Updates on the Management of Severe Acute Malnutrition in Infants and Children. Geneva: World Health Organization; 2013. PMID: 24649519. |
|
Reviewer 2 Comment # 09 Line 70: 'Systematic infection (or systemic?)' is a complication of SAM and therefore it does not pertain to uncomplicated SAM. |
The authors have acknowledged and complied with the reviewer’s comments |
|
Reviewer 2 Comment # 10 Table 2: 'Pair' is not needed. |
The authors have acknowledged the reviewer’s comments and believe that the pre- and post-comparison of the growth characteristics of children with uncomplicated SAM in the therapy group (RUTF plus prebiotics) produced some intriguing findings, which are shown in Table 2. The significant distinctions between the outcome measures before and after this study's intervention need to be emphasized. |
Reviewer 3 Report
The manuscript titled "Double-blind parallel treatment randomized controlled trial of prebiotics efficacy for severe acute malnutrition children in Southern Punjab, Pakistan" aimed to determine the effect of prebiotics supplementation with ready-to-use therapeutic use (RUTF) on malnourished children below the age of 5 years in a double blinded trial. The manuscript is well written, scientifically sound and all the methods are described well in detail. The authors found that prebiotics supplementation is effective and efficient in improving the effect of RUTF as determined by body weight gain and other improved hematological parameters.
Author Response
> The manuscript titled "Double-blind parallel treatment randomized controlled trial of prebiotics efficacy for severe acute malnutrition children in Southern Punjab, Pakistan" aimed to determine the effect of prebiotics supplementation with ready-to-use therapeutic use (RUTF) on malnourished children below the age of 5 years in a double blinded trial. The manuscript is well written, scientifically sound and all the methods are described well in detail. The authors found that prebiotics supplementation is effective and efficient in improving the effect of RUTF as determined by body weight gain and other improved hematological parameters.
The authors have acknowledged the reviewer’s comments.